# Long Non-Coding RNAs in Cardiac and Pulmonary Fibroblasts and Fibrosis

**DOI:** 10.3390/ncrna8040053

**Published:** 2022-07-15

**Authors:** Mirolyuba Ilieva, Shizuka Uchida

**Affiliations:** Center for RNA Medicine, Department of Clinical Medicine, Aalborg University, DK-2450 Copenhagen SV, Denmark; mirolyubasi@dcm.aau.dk

**Keywords:** cardiac, fibroblast, fibrosis, heart, lncRNA, lung

## Abstract

The cardiopulmonary system delivers oxygen throughout the body via blood circulation. It is an essential part of the body to sustain the lives of organisms. The integral parts of the cardiopulmonary system—the heart and lungs—are constantly exposed to damaging agents (e.g., dust, viruses), and can be greatly affected by injuries caused by dysfunction in tissues (e.g., myocardial infarction). When damaged, mesenchymal cells, such as fibroblasts, are activated to become myofibroblasts to initiate fibrosis as part of a regenerative mechanism. In diseased states, the excess accumulation of extracellular matrices secreted by myofibroblasts results in further dysfunction in the damaged organs. These fibrotic tissues cannot easily be removed. Thus, there is a growing interest in understanding the fibrotic process, as well as finding biomolecules that can be targets for slowing down or potentially stopping fibrosis. Among these biomolecules, the interest in studying long non-coding RNAs (lncRNAs; any non-protein-coding RNAs longer than 200 nucleotides) has intensified in recent years. In this commentary, we summarize the current status of lncRNA research in the cardiopulmonary system by focusing on cardiac and pulmonary fibrosis.

## 1. Introduction

Long non-coding RNA (lncRNA) is a collective term used to describe any non-protein-coding RNAs whose lengths are longer than 200 nucleotides. Compared to protein-coding mRNAs, lncRNAs’ expression is cell-type-specific, although at lower expression levels than mRNAs [1,2]. The exact number of lncRNAs is currently unknown, but it is predicted to be higher than that of mRNAs [3,4]. Given that there are many functions of proteins, the functions of lncRNAs are diverse, ranging from molecular scaffolds for epigenetic and transcription factor complexes, to decoys for other macromolecules (e.g., DNA, RNA, and proteins), signaling molecules, and regulators of mRNA stability and translation [5,6,7].

Fibrosis is a common pathological process in chronic inflammatory diseases, including cardiovascular and lung diseases [8,9]. During fibrosis, fibroblasts are activated to become myofibroblasts to deposit excess extracellular matrices consisting of macromolecules, such as collagens, glycoproteins, and matrix metalloproteinases. As a part of the wound-healing process in some cases (e.g., burns, skin wounds), fibrosis is a necessary process for tissue regeneration [10,11]. However, in other cases, where the complete regeneration of damaged tissue is not possible (e.g., heart, lungs), pathological fibrosis leads to the dysfunction of organs, resulting in various diseases [12,13]. As it is not possible to remove the fibrotic cells without damaging the surrounding tissues, early detection of pathological fibrosis is important to slow down or possibly stop irreversible fibrosis. In this regard, intensive research has been conducted to identify genes, proteins, and signaling pathways activated during fibrosis. Among these macromolecules and signaling pathways, lncRNAs are increasingly being investigated as a missing link to understand fibrosis in various fibrotic diseases [14,15,16,17,18,19,20]. In this commentary, we summarize the current status of lncRNA research in the cardiopulmonary system by focusing on cardiac and pulmonary fibrosis.

## 2. LncRNAs in Cardiac Fibrosis

Most cardiovascular diseases (e.g., myocardial infarction, cardiac hypertrophy) result in cardiac fibrosis, where cardiac fibroblasts are activated to become myofibroblasts to deposit excess extracellular matrices [21,22]. As it is not currently possible to replace fibrotic scarred tissues in the heart, the research on cardiac fibrosis has intensified in recent years. Through high-throughput screening via microarrays and RNA sequencing (RNA-Seq), a number of lncRNAs have been identified as being involved in cardiac fibrosis [23,24,25] (Figure 1A). Among them, the first functionally and mechanistically characterized cardiac fibrotic lncRNA is *Chrf* (cardiac hypertrophy-related factor), whose function is discussed below. This previous study was followed by several functional and mechanistic lncRNA studies, including *H19* (H19 imprinted maternally expressed transcript) acting to antagonize the suppressor of collagen 1A1, Ybx1 (Y box protein 1), by directly binding it under hypoxia [26]; *Meg3* (maternally expressed 3) interacting with P53 to regulate the expression of *Mmp2* (matrix metallopeptidase 2) [27]; and *Wisper* (Wisp2 super-enhancer-associated RNA) interacting with Tial1 (Tia1 cytotoxic granule-associated RNA-binding protein-like 1, also known as TIAR) to regulate alternative splicing of *Plod2* (procollagen lysine, 2-oxoglutarate 5-dioxygenase 2) to stabilize the extracellular matrix [23]. These lncRNA studies, and more recent ones, are summarized in Table 1.

As exemplified by the lncRNA *HOTAIR* (HOX transcript antisense RNA) [31], it was previously popular to investigate the function of lncRNAs as molecular scaffolds of histone modifiers—especially binding to the core PRC2 (polycomb repressive complex 2) component EZH2 (enhancer of zeste 2 polycomb repressive complex 2 subunit). However, after the series of studies reporting that EZH2 promiscuously binds any RNAs [32,33,34,35,36], including lncRNAs, many researchers have become more careful about reporting lncRNAs as molecular scaffolds for histone modifiers. Instead, the current trend is to understand the function of lncRNAs as miRNA (microRNA) sponges. MicroRNAs are regulatory RNA species with an average length of 22 nucleotides for their mature forms. The primary function of miRNAs is to bind the 3′-UTR (untranslated region) of mRNAs to negatively regulate their gene expression by degrading their target mRNAs and/or by silencing translation. Partially fueled by the established techniques available to study miRNAs (e.g., gain-of-function via miRNA mimics, loss-of-function via anti-miR (e.g., GapmeRs), and binding assay with the mutation of miRNA-binding sites in the 3′-UTR of the target mRNA and readout by luciferase assay), many lncRNAs are reported to bind miRNAs, including those of cardiac fibrotic lncRNAs. For example, the lncRNA *Chrf* sequesters *miR-489*, which targets *Myd88* (MYD88 innate immune signal transduction adaptor) in cardiomyocytes to regulate cardiac hypertrophy [37].

Since the discovery of miRNAs in Caenorhabditis elegans in 1993 [38], miRNA research has intensified in recent years. This has led to the usage of miRNAs as diagnostic biomarkers and potential therapeutic targets for various diseases, including cardiovascular disease [39,40,41]. In the context of fibrosis, the specific term fibromiR is used to describe the group of miRNAs involved in fibrosis, including *miR-1*, *miR-21*, *miR-29*, and *miR-208* [42]. In cardiac fibrosis, several lncRNAs have been reported to function as miRNA sponges to sequester fibromiRs (Figure 1B). For example, the lncRNA *H19* sequesters *miR-29a/b-3p*, which targets *Vegfa* (vascular endothelial growth factor A) to promote the proliferation of rat cardiac fibroblasts and collagen synthesis via the TGF-β signaling pathway in vitro [43]. Interestingly, the *miR-29* family members [44] are also shown to be sequestered by another lncRNA—*TUG1* (taurine upregulated 1). In cardiac fibroblasts, *TUG1* sequesters *miR-29c* to promote the activation of murine cardiac fibroblasts to myofibroblasts under hypoxic conditions in vitro [45]. Another study showed that *TUG1* sequesters *miR-29b-3p*, which targets *TGFB1* (transforming growth factor beta 1) to regulate the proliferation of human cardiac fibroblasts treated with angiotensin II in vitro [46]. Given that fibrosis is a general disease mechanism for chronic inflammation, more fibromiRs will be shown to be sequestered by lncRNAs in cardiac fibroblasts and fibrosis in the near future.

Although the mechanism of lncRNAs as miRNA sponges is an attractive function, it is not clear how generally lowly expressed lncRNAs (except for those highly expressed lncRNAs, such as *MALAT1* (metastasis associated lung adenocarcinoma transcript 1)) sequester much more abundantly expressed miRNAs. Moreover, many miRNAs bind more than one site of the 3′-UTR of protein-coding genes for gene regulation. Given that most mRNAs are much more highly expressed than lncRNAs, it is not clear how efficiently each lncRNA functions as an miRNA sponge in a cell—especially when the copy number of the lncRNA in a cell has not been reported in the original studies, along with the copy number of the miRNA that the target lncRNA sequences. To complicate the matter further, individual lncRNAs have been reported to sponge several miRNAs, as in the case of the lncRNA *GAS5* (growth arrest specific 5), for *miR-21*, *miR-26a/b-5p*, and *miR-5325p* in cardiac fibroblasts and fibrosis [47,48,49,50]. Another example is *TUG1*, which sequesters *miR-29b-3p*, *miR-29c*, *miR-133b*, and *miR-590* [45,46,51,52]. As there are more miRNAs being identified to bind these two exemplified lncRNAs in other cell types (e.g., cardiomyocytes, endothelial cells), cellular statuses, and diseases [51,52,53,54,55,56,57], it is important to perform a comparative study of lncRNAs and several miRNAs being reported to be sequestered by the target lncRNAs by measuring the copy numbers of each RNA species. Without such a head-to-head comparative study, more and more miRNAs will be reported to be sequestered by one lncRNA, despite its expression level being far lower than those of the sequestered miRNAs.

## 3. LncRNAs in Pulmonary Fibrosis

Pulmonary fibrosis is a generalized term used to describe chronic and progressive interstitial lung diseases, including sarcoidosis, Langerhans-cell granulomatosis, eosinophilic pneumonia, lymphangioleiomyomatosis, and pulmonary alveolar proteinosis [58]. Among them, idiopathic pulmonary fibrosis (IPF) is the most common form. Unlike cardiac fibrosis, where cardiac fibroblasts are the major source of myofibroblasts, the origin of myofibroblasts in the lungs is still debated, including alveolar epithelial cells, circulating fibrocytes, and lung stromal cell subpopulations (e.g., resident fibroblasts, pericytes, and resident mesenchymal stem cells) [59]. Thus, the lncRNA studies of pulmonary fibrosis extend far beyond those of fibroblasts, complicating the research itself, as many lncRNAs are expressed in a cell-type-specific manner. Generally speaking, not only the activation of fibroblasts, but also epithelial-to-mesenchymal transition (EMT)—the process in which differentiated epithelial cells acquire mesenchymal phenotypes [60]—is investigated for pulmonary fibrotic lncRNAs (Figure 2A). Furthermore, unlike cardiac fibrosis, some studies of pulmonary fibrosis involve environmental and occupational agents (e.g., asbestos, cigarette smoke), as these are known causes of pulmonary fibrosis—especially IPF. In addition, the most frequently used murine injury model of IPF is the injection of the chemotherapy agent bleomycin, which has pulmonary toxicity [61,62]. For example, transgenic mice overexpressing the lncRNA *PFI* (pulmonary fibrosis inhibitor, also known as *NONMMUT060091*) were challenged with intratracheal injection of bleomycin to induce pulmonary fibrosis. Molecular and histological analyses showed that overexpression of *PFI* alleviated the negative effects of bleomycin by reducing the deposition of extracellular matrix and differentiation of myofibroblasts [63]. Mechanistically, *PFI* binds SRSF1 (serine- and arginine-rich splicing factor 1) to inhibit its function on alternative splicing. Other pulmonary fibrotic lncRNAs are summarized in Table 2.

As was the case for cardiac fibrotic lncRNAs, the function as miRNA sponges is a popular mechanism investigated for pulmonary fibrotic lncRNAs. Unsurprisingly, many lncRNAs identified as miRNA sponges in cardiac fibrosis are also implicated in pulmonary fibrosis (Figure 2B). Thus, rather than simply listing all lncRNAs functioning as miRNA sponges in pulmonary fibrosis, here, the same lncRNAs identified as miRNA sponges in both cardiac and pulmonary fibrosis are discussed. For example, *H19* sequesters *miR-455* and *miR-22-3p*, which target *CCN2* (cellular communication network factor 2, also known as *CTGF*) [70] and *Kdm3a* (lysine (K)-specific demethylase 3A) mRNAs [71], respectively, in cardiac fibrosis. In pulmonary fibrosis, *H19* binds *miR-29b*, *miR-423-5p*, and *miR-196a* to silence the translation of *Col1a1* (collagen, type I, alpha 1) [72], *Foxa1* (forkhead box A1) [73], and *COL1A1* mRNAs [74], respectively. Other lncRNAs functioning as miRNA sponges in both cardiac and pulmonary fibrosis are summarized in Table 3.

As stated in the previous section, if indeed lncRNAs primarily function as miRNA sponges, it is a very complicated cascade of events, as the same miRNA is sequestered by several lncRNAs, while one miRNA has several hundred predicted mRNA targets. In this regard, it is interesting to note that the lncRNA *FENDRR* (FOXF1-adjacent non-coding developmental regulatory RNA) sequesters *miR-214* while interacting with ACO1 (aconitase 1, also known as IRP1) to decrease cellular iron concentration and reduce pulmonary fibrosis [66]. Another example is the exosomal lncRNA *HOTAIRM1* (HOXA transcript antisense RNA, myeloid-specific 1), which sequesters *miR-30d-3p* while recruiting YY1 (YY1 transcription factor) to upregulate *HSF1* (heat shock transcription factor 1) to promote extracellular matrix remodeling [67]. Although these two studies report lncRNAs as miRNA sponges, other mechanisms of action are proposed to tease out the exact mechanisms of action of lncRNAs.

## 4. Discussions

For lncRNAs to be recognized further as important molecules for signaling pathways, it is of utmost importance that each lncRNA study carefully characterize expression patterns and functions, including those of isoforms (transcripts of each lncRNA gene). The information about known isoforms can be examined easily from the public databases, such as NCBI’s Entrez Gene and Ensembl. Without such careful studies, it will be a long time before lncRNAs can be considered as potential diagnostic biomarkers of various diseases—especially fibrosis—and possibly as druggable targets. As for fibrotic lncRNAs, especially in the cardiopulmonary system and its associated diseases, the number of functionally characterized fibrotic lncRNAs is scarce. To assist in the identification of lncRNAs expressed in fibroblasts and involved in fibrosis—especially in cardiac and pulmonary fibrosis—we recently released the web application FibroDB [86] (https://rnamedicine.shinyapps.io/FibroDB/, accessed on 12 July 2022).

The identification of lncRNAs involved in cardiac and pulmonary fibrosis is only the first step. The identified lncRNAs must be characterized carefully by performing experiments, including 5/*3*′-RACE (rapid amplification of cDNA ends) and Northern blotting to identify isoforms of the target lncRNA; in vitro transcription and translation assays to test the coding potential of the target lncRNA; RT-PCR (reverse transcription polymerase chain reaction) using nuclear and cytosol fractions of the target cell separately and FISH (fluorescent in situ hybridization) to identify the subcellular localization of the target lncRNA; gain/loss-of-function experiments to observe phenotypic changes; RNA pull-down followed by mass spectrometry or more advanced methods (e.g., CHART (capture hybridization analysis of RNA targets), ChIRP (chromatin isolation by RNA purification), and RAP (RNA affinity purification), which are comprehensively reviewed in [87]) to identify potential protein binding partners of the target lncRNA; and RIP-PCR (RNA immunoprecipitation followed by RT-PCR) to verify the binding between the target lncRNA and its protein-binding partner(s). It is unfortunate that most lncRNA studies published to date fail to report the presence (or absence) of isoforms, as many isoforms of the target lncRNAs have distinctive tissue/cell-specific expression patterns, as in the case of the paternal imprinting lncRNA gene *Airn* [88]. On top of this, as more and more lncRNAs are being discovered due to the results of the readily available RNA-Seq method, public databases (e.g., NCBI, Ensembl) have been updating the annotations of human transcriptomes and other model organisms at a rapid pace. This has resulted in significant changes in the numbers of lncRNA genes and their isoforms. Thus, it is highly recommended to check the primer sequences for the target lncRNA regularly (and, of course, before publication) to make sure that the predicted PCR product amplifies the product at the expected product size, and no other products at various sizes. This can be easily performed using the UCSC In-Silico PCR tool (https://genome.ucsc.edu/cgi-bin/hgPcr, accessed on 12 July 2022).

## 5. Conclusions

As summarized in this commentary, a number of lncRNAs involved in cardiac and pulmonary fibrosis have been identified and studied. The most common mechanism to date for these fibrotic lncRNAs is as miRNA sponges, although there remains the question of the amounts of lncRNAs available to sequester much more abundant miRNAs, and why some lncRNAs (e.g., *GAS5*, *H19*, *MALAT1*, *MIAT*, *PVT1*) have several miRNAs binding to one lncRNA. These questions should be answered by further experiments, and this mini-review should stimulate such mechanistic studies in cardiac and pulmonary fibrosis.

## Figures and Tables

**Figure 1 ncrna-08-00053-f001:**
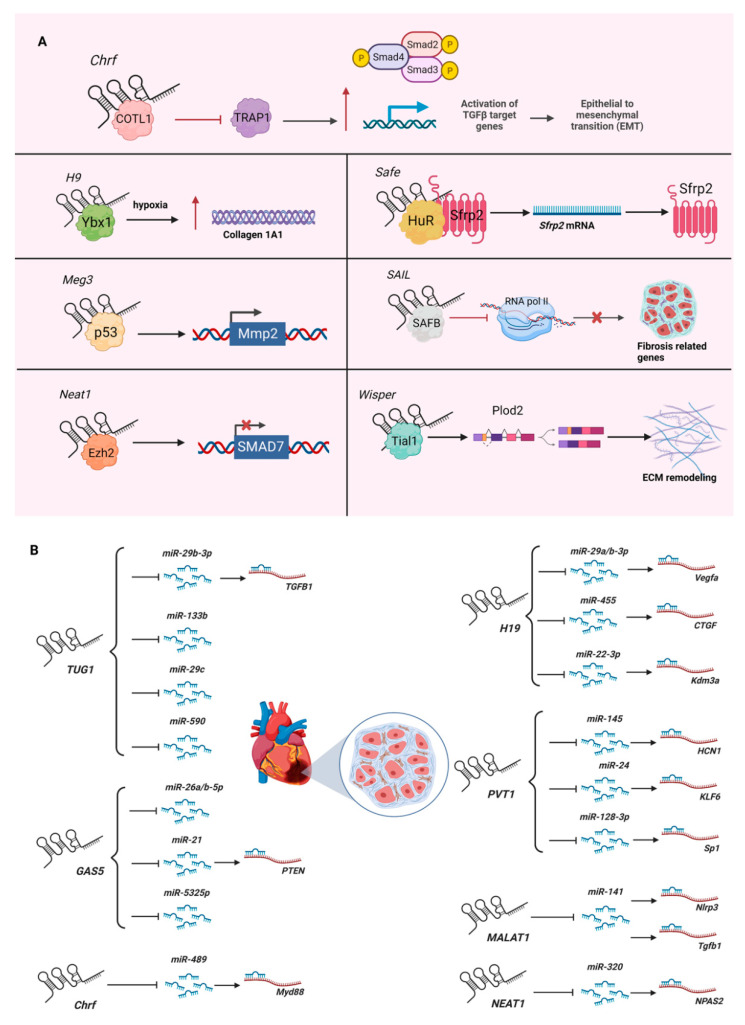
Role of lncRNAs in cardiac fibrosis: (**A**) Mechanisms of action of lncRNAs in the pathogenesis of cardiac fibrosis. (**B**) LncRNAs as miRNA sponges for cardiac fibromiRs. Please see Table 1 for mechanistic descriptions. Figure created with BioRender.com, accessed on 22 June 2022.

**Figure 2 ncrna-08-00053-f002:**
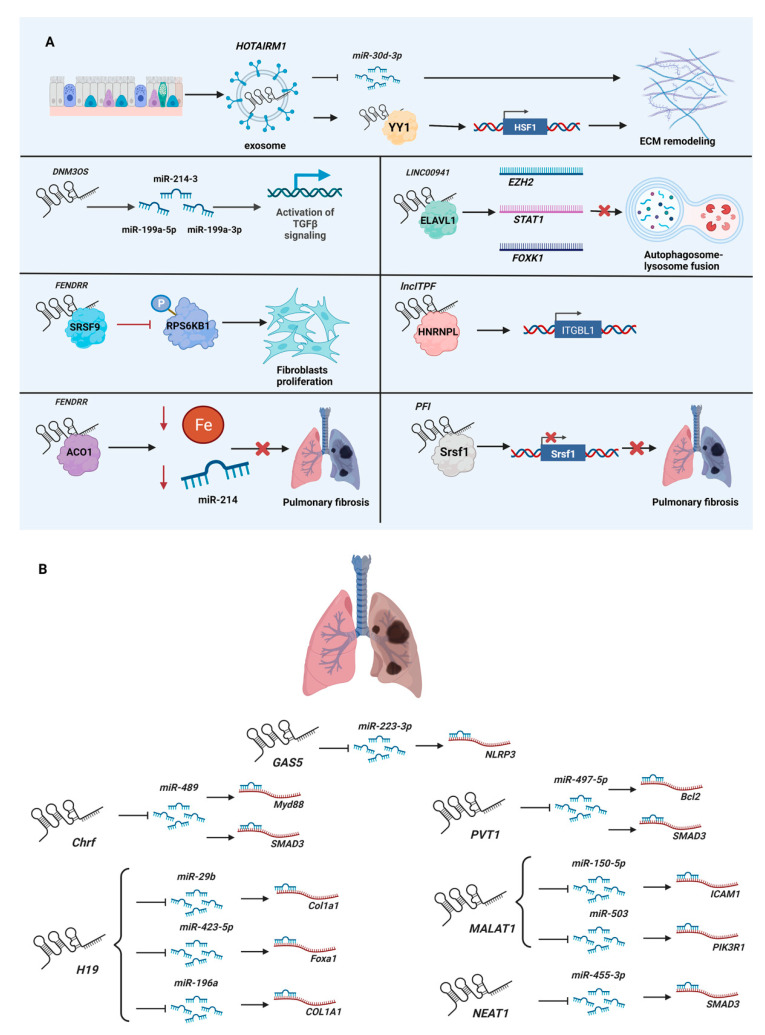
Role of lncRNAs in pulmonary fibrosis: (**A**) Mechanisms of action of lncRNAs in the pathogenesis of pulmonary fibrosis. (**B**) LncRNAs as miRNA sponges for pulmonary fibromiRs. Please see Table 2 for mechanistic descriptions. Figure created with BioRender.com, accessed on 22 June 2022.

**Table 1 ncrna-08-00053-t001:** List of functionally and mechanistically characterized cardiac fibrotic lncRNAs.

Name of LncRNA	Study Model	Function	Reference
*Cfast* (cardiac fibroblast-associated transcript; also known as *AK048087*)	Knockdown by siRNA in primary neonatal murine cardiac fibroblasts; knockdown by lentiviral shRNA in infarcted (permanent ligation of left anterior descending coronary artery) or hypertrophied murine hearts (injection of isoproterenol).	Binds COTL1 (coactosin-like F-actin-binding protein 1) to competitively inhibit its interaction with TRAP1 (TNF receptor-associated protein 1), which enhances TGF-β signaling by augmenting SMAD2/SMAD4 complex formation.	[24]
*H19* (H19 imprinted maternally expressed transcript)	Overexpression by AAV9 (adeno-associated virus serotype 9) in infarcted murine hearts (permanent ligation of left anterior descending coronary artery) or intraperitoneal injection into postnatal 8–12-day-old mice; myocardial infarction in *H19*-knockout mice generated using the CRISPR/Cas9 system; knockdown by siRNA or overexpression in primary adult and neonatal murine cardiac fibroblasts, hiPSC-CFs (human-induced-pluripotent-stem-cell-derived cardiac fibroblasts), and NIH3T3 cells.	Binds Ybx1 (Y box protein 1) under hypoxia to cause de-repression of collagen 1A1 expression.	[26]
*Meg3* (maternally expressed 3)	Knockdown by GapmeR in primary adult murine cardiac fibroblasts or hypertrophied (transverse aortic constriction) murine hearts.	Interacts with P53 to regulate the expression of *Mmp2* (matrix metallopeptidase 2).	[27]
*Neat1* (nuclear paraspeckle assembly transcript 1)	Knockdown by siRNA or overexpression by adenovirus in primary neonatal murine cardiac fibroblasts treated with TGF-β1.	Binds Ezh2 to recruit it to the promoter of *Smad7* to inhibit its expression.	[28]
*Safe* (*AK137033*)	Knockdown by shRNA or CRISPR/Cas9-mediated knockout in primary adult murine cardiac fibroblasts treated with TGF-β; knockdown by lentiviral shRNA or overexpression by lentivirus in infarcted murine hearts (permanent ligation of left anterior descending coronary artery).	Promotes the *Safe*-*Sfrp2*-HuR (Elavl1, ELAV (embryonic lethal, abnormal vision)-like 1 (Hu antigen R)) complex-mediated *Sfrp2* (secreted frizzled related protein 2) mRNA stability and protein expression.	[29]
*SAIL* (scaffold attachment factor B interacting lncRNA; *Gm19522* (predicted gene, 19522))	Knockdown by siRNA or overexpression by plasmids in primary neonatal murine cardiac fibroblasts or human cardiac fibroblasts treated with TGF-β1; overexpression by Ad5 (adenovirus serotype 5) in infarcted murine hearts (permanent ligation of left anterior descending coronary artery).	Binds with SAFB (scaffold attachment factor B) to block its access to RNA pol II (RNA polymerase II), and reduces the transcription of fibrosis-related genes.	[30]
*Wisper* (Wisp2 super-enhancer-associated RNA)	Knockdown by GapmeR in primary neonatal and adult murine cardiac fibroblasts, primary adult murine lung fibroblasts, primary human cardiac fibroblasts, and infarcted murine hearts (permanent ligation of left anterior descending coronary artery); overexpression by CRISPR-on in P19CL6 cells.	Interacts with Tial1 (Tia1 cytotoxic granule-associated RNA-binding protein-like 1, also known as TIAR) to regulate alternative splicing of *Plod2* (procollagen lysine, 2-oxoglutarate 5-dioxygenase 2) to stabilize the extracellular matrix.	[23]

**Table 2 ncrna-08-00053-t002:** List of functionally and mechanistically characterized pulmonary fibrotic lncRNAs.

Name of LncRNA	Study Model	Function	Reference
*DNM3OS* (DNM3 opposite strand/antisense RNA)	Knockdown by GapmeR in the human lung fibroblastic cell line MRC5 and a murine bleomycin-induced lung fibrosis model.	Encodes three fibromiRs (*miR-199a-5p/3p* and *miR-214-3*), which regulate SMAD and non-SMAD components of TGF-β signaling.	[64]
*FENDRR* (FOXF1 adjacent non-coding developmental regulatory RNA)	Knockdown by lentiviral shRNA in human lung fibroblastic cell line LL29; overexpression by adenovirus in a murine asbestos-induced lung fibrosis model.	Binds SRSF9 (serine- and arginine-rich splicing factor 9) to inhibit the phosphorylation of RPS6KB1 (ribosomal protein S6 kinase B1, also known as PS6K), thereby suppressing the proliferation of fibroblasts.	[65]
*FENDRR* (FOXF1 adjacent non-coding developmental regulatory RNA)	Knockdown by lentiviral shRNA in the human lung fibroblastic cell lines HFL1 and LL29; overexpression by adenovirus in a murine bleomycin-induced lung fibrosis model.	Interacts with ACO1 (aconitase 1, also known as IRP1) to decrease cellular iron concentration and sequester pro-fibrotic *miR-214* to reduce pulmonary fibrosis.	[66]
*HOTAIRM1* (HOXA transcript antisense RNA, myeloid-specific 1)	Exosomes isolated from the murine lung epithelial cell line MLE-12 and co-cultured with primary murine lung fibroblasts; overexpressed by lentivirus in the murine lung epithelial cell line MLE-12, extracted exosomes, and then injected exosomes and lentivirus into a murine bleomycin-induced lung fibrosis model.	Released via exosomes from alveolar epithelial cells to lung fibroblasts, where it sequesters *miR-30d-3p* and recruits YY1 (YY1 transcription factor) to upregulate HSF1 (heat shock transcription factor 1), thereby promoting extracellular matrix remodeling.	[67]
*LINC00941* (long intergenic non-protein coding RNA 941, also known as *lncIAPF*)	Knockdown by siRNA and overexpression by adenovirus in the human lung fibroblast cell line MRC-5; overexpression by adenovirus in the murine fibroblast cell line L929; overexpression by adenovirus in a murine bleomycin-induced lung fibrosis model.	Formed an RNA–protein complex with ELAVL1 to inhibit autophagosome fusion with a lysosome by controlling the stability of *EZH2*, *STAT1* (signal transducer and activator of transcription 1), and *FOXK1* (forkhead box K1) mRNAs.	[68]
*lncITPF* (lncRNA regulates its host gene Itgbl1 during pulmonary fibrogenesis, also known as *MRAK053938*)	Knockdown by siRNA and overexpression in the human lung fibroblast cell line MRC-5; knockdown by lentiviral shRNA in a murine bleomycin-induced lung fibrosis model.	Binds HNRNPL (heterogeneous nuclear ribonucleoprotein L) to epigenetically regulate its host gene, *ITGBL1* (integrin subunit beta-like 1).	[69]
*PFI* (pulmonary fibrosis inhibitor, also known as *NONMMUT060091*)	Knockdown by LncRNA Smart Silencer and overexpression in primary murine lung fibroblasts; PFI transgenic (TG-PFI) mice with a murine bleomycin-induced lung fibrosis model.	Binds Srsf1 (serine and arginine-rich splicing factor 1) to repress its expression and pro-fibrotic activity.	[63]

**Table 3 ncrna-08-00053-t003:** List of cardiac and pulmonary fibrotic lncRNAs functioning as miRNA sponges: In the Cardiac Fibrosis and Pulmonary Fibrosis columns, miRNAs and their target mRNAs are shown, which are separated by a forward slash. When there are multiple entries (i.e., miRNA/target mRNA) for each column, each entry is separated by a semicolon. This applies to the corresponding references as well.

Name of lncRNA	Cardiac Fibrosis	References	Pulmonary Fibrosis	References
*Chrf* (cardiac hypertrophy-related factor)	*miR-489*/*Myd88*	[37]	*miR-489*/*Myd88* & *Smad3*	[75]
*GAS5* (growth arrest specific 5)	*miR-21*/*PTEN*	[50]	*miR-223-3p*/*NLRP3*	[76]
*H19* (H19 imprinted maternally expressed transcript)	*miR-455*/*CTGF*; *miRNA-22-3p*/*Kdm3a*	[70]; [71]	*miR-29b*/*Col1a1*; *miR-423-5p*/*Foxa1*; *miR-196a*/*COL1A1*	[72]; [73]; [74]
*MALAT1* (metastasis-associated lung adenocarcinoma transcript 1)	*miR-141*/*Nlrp3* & *Tgfb1*	[77]	*miR-150-5p*/*ICAM1*; *miR-503*/*PIK3R1*	[78]; [79]
*NEAT1* (nuclear paraspeckle assembly transcript 1)	*miR-320*/*NPAS2*	[80]	*miR-455-3p*/*SMAD3*	[81]
*PVT1* (Pvt1 oncogene	*miR-145*/*HCN1*; *miR-24*/*KLF6*; *miR-128-3p*/*Sp1*	[82]; [83]; [84]	*miR-497-5p*/*Bcl2* & *Smad3*	[85]

## Data Availability

Not applicable.

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
