# Peer review of "Long Non-Coding RNAs in Cardiac and Pulmonary Fibroblasts and Fibrosis"

_ncrna, 2022, doi:10.3390/ncrna8040053_

Round 1

Reviewer 1 Report

The authors have presented a succinct summary of recent papers describing the roles of various lncRNAs in cardiac and pulmonary fibrosis. The presentation is primarily as an overview and listing of recent publications and their major findings without a clear synthesis of these data into any overarching theme or take-away message. While the topic is timely and the summary of the recent research is good, I think the commentary would be more effective if there were a few key take-away messages for the manuscript.

In addition, I would recommend the following:

- Would change the title so as not to conflate cardiac and pulmonary fibroblasts and fibrosis. Would suggest, "Long non-coding RNAs in cardiac and pulmonary fibroblasts and fibrosis".

- There are many grammatical errors related to missing articles, subject-verb agreement, mis-spellings such that the manuscript would benefit from professional copy editing.

- Since the tables describe the mechanisms illustrated by the figures, would reference the appropriate table in the figure legend (i.e., In the caption of Figure 1, state something along the lines of "Please see Table 1 for mechanistic descriptions.")

- The tables are very hard to read. I would suggest vertically aligning each cell to the top and left aligning all of the text, rather than centering. Would also add extra spacing between each line. Currently all of the text visually blurs together. 

Author Response

The authors have presented a succinct summary of recent papers describing the roles of various lncRNAs in cardiac and pulmonary fibrosis. The presentation is primarily as an overview and listing of recent publications and their major findings without a clear synthesis of these data into any overarching theme or take-away message. While the topic is timely and the summary of the recent research is good, I think the commentary would be more effective if there were a few key take-away messages for the manuscript.

Response: Thank you very much for your valuable comments. The manuscript has been extensively modified to include more take-away messages.

In addition, I would recommend the following:

- Would change the title so as not to conflate cardiac and pulmonary fibroblasts and fibrosis. Would suggest, "Long non-coding RNAs in cardiac and pulmonary fibroblasts and fibrosis".

Response: The title has been changed as suggested.

- There are many grammatical errors related to missing articles, subject-verb agreement, mis-spellings such that the manuscript would benefit from professional copy editing.

Response: The manuscript has been read extensively to correct grammatical errors.

- Since the tables describe the mechanisms illustrated by the figures, would reference the appropriate table in the figure legend (i.e., In the caption of Figure 1, state something along the lines of "Please see Table 1 for mechanistic descriptions.")

Response: The suggested text was added to each figure legend.

- The tables are very hard to read. I would suggest vertically aligning each cell to the top and left aligning all of the text, rather than centering. Would also add extra spacing between each line. Currently all of the text visually blurs together.

Response: The horizontal lines were added to tables to improve the visibility and readability of the tables. Furthermore, texts within each cell were vertically aligned to the top and left aligned.

Reviewer 2 Report

1. Tables must be improved. The addition of lines to distinguish individual items is helpful.
2. When discussing individual lncRNA, it would be more informative by mentioning in which cellular compartments a lncRNA is localized and where this lncRNA plays major roles.

Author Response

  1. Tables must be improved. The addition of lines to distinguish individual items is helpful.

Response: Thank you very much for your valuable comments. The horizontal lines were added to tables to improve the visibility and readability of the tables. Furthermore, texts within each cell were vertically aligned to the top and left aligned.

  1. When discussing individual lncRNA, it would be more informative by mentioning in which cellular compartments a lncRNA is localized and where this lncRNA plays major roles.

Response: Instead of making the main text longer to read for the readers, the information about cellular compartments as well as animal model used are summarized in tables.

Round 2

Reviewer 1 Report

The authors have addressed my concerns.